# Laminar specificity and coverage of viral-mediated gene expression restricted to GABAergic interneurons and their parvalbumin subclass in marmoset primary visual cortex

**Frederick Federer[1], Justin Balsor[1], Alexander Ingold[1†], David P Babcock[1‡], Jordane Dimidschstein[2], Alessandra Angelucci[1*]**

[1]Department of Ophthalmology and Visual Science, Moran Eye Institute, University of Utah, Salt Lake City, United States; [2]Regel Therapeutics, Boston, United States

**\*For correspondence:**
alessandra.angelucci@hsc.utah.edu

**Present address:** [†]Department of Electrical Engineering and Computer Science, University of Utah, Salt Lake City, United States; [‡]Stritch School of Medicine, Loyola University, Chicago, United States

**Competing interest:** The authors declare that no competing interests exist.

**Abstract** In the mammalian neocortex, inhibition is important for dynamically balancing excitation and shaping the response properties of cells and circuits. The various computational functions of inhibition are thought to be mediated by different inhibitory neuron types, of which a large diversity exists in several species. Current understanding of the function and connectivity of distinct inhibitory neuron types has mainly derived from studies in transgenic mice. However, it is unknown whether knowledge gained from mouse studies applies to the non-human primate, the model system closest to humans. The lack of viral tools to selectively access inhibitory neuron types has been a major impediment to studying their function in the primate. Here, we have thoroughly validated and characterized several recently developed viral vectors designed to restrict transgene expression to GABAergic cells or their parvalbumin (PV) subtype, and identified two types that show high specificity and efficiency in marmoset V1. We show that in marmoset V1, AAV-h56D induces transgene expression in GABAergic cells with up to 91–94% specificity and 79% efficiency, but this depends on viral serotype and cortical layer. AAV-PHP.eB-S5E2 induces transgene expression in PV cells across all cortical layers with up to 98% specificity and 86–90% efficiency, depending on layer. Thus, these viral vectors are promising tools for studying GABA and PV cell function and connectivity in the primate cortex.

## eLife assessment

Unlocking the potential of molecular genetic tools (optogenetics, chemogenetics, sensors, etc.) for the study of systems neuroscience in nonhuman primates requires the development of effective regulatory elements for cell-type-specific expression to facilitate circuit dissection. This study provides a **valuable** building block by carefully characterizing the laminar expression profile of two optogenetic enhancers, one designed for general GABA+ergic neurons (h56D) and the second (S5E2) for parvalbumin+ cell-type selective expression in the marmoset primary visual cortex. This study contributes **solid** evidence to our understanding of these tools but is limited by the understandably small number of animals used.

## Introduction

The computations performed by the neocortex result from the activity of neural circuits composed of glutamatergic excitatory and GABAergic inhibitory neurons. Although representing only 15–30% of all cortical neurons, inhibitory neurons profoundly influence cortical computations and cortical dynamics. For example, they influence how excitatory neurons integrate information, shape neuronal tuning properties, modulate neuronal responses based on sensory context and behavioral state, and maintain an appropriate dynamic range of cortical excitation (*Ferster and Miller, 2000*; *Shapley et al., 2003*; *Tremblay et al., 2016*). These various functions of inhibition are thought to be mediated by different inhibitory neuron types, of which a large diversity has been identified in several species, each having distinct chemical, electrophysiological, and morphological properties (*Ascoli et al., 2008*; *Burkhalter, 2008*; *Kubota et al., 2016*).

In mouse cortex, the expression of specific molecular markers identifies three major, largely non-overlapping classes of inhibitory neurons: parvalbumin- (PV), somatostatin- (SOM), and serotonin receptor (5HT3aR, a larger class which includes vasoactive intestinal peptide or VIP cells)-expressing neurons (*Xu et al., 2010*; *Rudy et al., 2011*). The creation of mouse lines selectively expressing Cre-recombinase in specific inhibitory neuron classes has led to a multitude of studies on the connectivity and function of each class (*Tremblay et al., 2016*; *Wood et al., 2017*; *Shin and Adesnik, 2023*). Distinct patterns of connectivity and function specific to each inhibitory neuron class are emerging from these mouse studies, but it remains unknown whether insights gained from mouse apply to inhibitory neurons in higher species such as primates. Understanding cortical inhibitory neuron function in the primate is critical for understanding cortical function and dysfunction in the model system closest to humans, where cortical inhibitory neuron dysfunction has been implicated in many neurological and psychiatric disorders, such as epilepsy, schizophrenia, and Alzheimer's disease (*Cheah et al., 2012*; *Verret et al., 2012*; *Mukherjee et al., 2019*).

A major impediment to studying inhibitory neuron function in primates has been the lack of tools for cell-type-specific expression of transgenes in this species. However, recent advances in the application to primates of cell-type-specific viral technology are beginning to enable studies of inhibitory neuron types in primate cortex. In particular, two recent studies have developed specific promoters or enhancers that restrict transgene expression from recombinant adeno-associated viral vectors (AAV) to GABAergic neurons, specifically the *mDlx* enhancer (*Dimidschstein et al., 2016*) and the *h56D* promoter (*Mehta et al., 2019*), in both rodents and primates. Viral strategies to restrict gene expression to PV neurons have also been recently developed (*Mehta et al., 2019*; *Vormstein-Schneider et al., 2020*; *Mich et al., 2021*).

To facilitate the application of these inhibitory-neuron specific viral vectors to studies of the primate cortex, we have performed a thorough validation and characterization of the laminar expression of reporter proteins mediated by several enhancer/promoter-specific AAVs. Here, we report results from the two vector types that have shown the greatest specificity of transgene expression in marmoset primary visual cortex (V1); specifically, we have tested three serotypes of the *h56D* promoter-AAV that restricts gene expression to GABAergic neurons (*Mehta et al., 2019*), and one serotype of the S5E2 enhancer-AAV that restricts gene expression to PV cells (*Vormstein-Schneider et al., 2020*). Using injections of these viral vectors in marmoset V1, combined with immunohistochemical identification of GABA and PV neurons, we find that the laminar distribution of reporter protein expression mediated by the GABA- and PV-enhancer AAVs validated in this study resembles the laminar distribution of GABA-immunoreactive (GABA+) and PV-immunoreactive (PV+) cells, respectively, in marmoset V1. Reporter protein expression mediated by the h56D-AAV is specific and robust, but the degree of specificity and coverage depended on serotype and cortical layer. We found that about 92% of PV cells in marmoset V1 are GABA+, and reporter protein expression mediated by the S5E2-AAV shows up to 98% specificity and 86–90% coverage, depending on layer. We conclude that these viral vectors offer the possibility of studying GABAergic and PV neuron connectivity and function in primate cortex.

## Results

We validated three serotypes (1,7,9) of a pAAV-h56D-tdTomato (*Mehta et al., 2019*) and the AAV-PHP.eB-S5E2.tdTomato (*Vormstein-Schneider et al., 2020*). We report results from 10 viral injections, of which 3 injections of AAV-h56D-tdTomato, and 7 injections of AAV-PHP.eB-S5E2.tdTomato, made

in 4 marmoset monkeys (see 'Materials and methods' and *Supplementary file 1*). Tissue sections through V1 were double immunoreacted for GABA and PV and imaged on a fluorescent microscope. We quantified the laminar distribution of viral-induced tdTomato (tdT) expression as well as of GABA+ and PV+ cells revealed by immunohistochemistry (IHC), and counted double- and triple-labeled cells to determine the specificity and coverage of viral-induced tdT expression across marmoset V1 layers (see 'Materials and methods').

## V1 laminar distribution of GABA+ and PV+ neurons

We first determined the V1 laminar distribution of GABA+ and PV+ neurons identified by IHC (*Figure 1*). To this goal, in each section used for analysis, we counted GABA+ and PV+ cells within 2 × 100-µm-wide regions of interest (ROIs) spanning all layers in dorsal V1 anterior to the posterior pole, for a total of six ROIs in three tissue sections selected randomly. Cortical layer boundaries were determined using DAPI and/or PV staining (*Figure 1B and C*), as we found that PV-IHC reveals laminar boundaries consistent with those defined by DAPI.

The laminar distribution of GABA+ and PV+ cells was quantified as percent of total GABA+ or PV+ cells (*Figure 2A*), as well as cell density (number of cells per unit area; *Figure 2B*), in each cortical layer. Both GABA+ and PV+ cell percent and density peaked in layers (L) 2/3 and 4C. There was no significant difference in the percent or density of GABA+ cells in L2/3 (33% ± 2, and 1294 cells/mm$^2$ ± 74.4, respectively) vs. L4C (33.4% ± 3.1, and 1052 cells/mm$^2$ ± 42.2, respectively), as determined by a Bonferroni-corrected Kruskal–Wallis test (p=1.00, n = 6 ROIs across three sections, L2/3 vs. L4C in *Figure 2A*) or by a Bonferroni-corrected ANOVA (p=1.00, n = 6 ROIs across three sections, L2/3 vs. L4C in *Figure 2B*). Similarly, the percent and density of PV+ cells in L4C (42.3% ± 3.6, and 888 cells/mm$^2$ ± 38.8, respectively) were not significantly different from those in L2/3 (31.3% ± 2.5 and 812 cells/mm$^2$ ± 65, respectively; p=1.00 for both comparisons, n = 6 ROIs). However, the density of PV+ cells in L4C was significantly higher than that in all remaining layers (p=0.028005 for 4C vs. 4A/B, and <0.001 for 4C vs. all other layers; ANOVA with Bonferroni correction; n = 6 ROIs), and L2/3 PV+ density was significantly higher than L1, 5 and 6 (p=5.93E$^{-11}$ for L2/3 vs. L1 and p=0.00037 L2/3 vs. L6, p=0.002 for L2/3 vs. L5, n = 6 ROIs), but not L4A/B (although the percent of PV+ cells in L2/3 was significantly higher than that in L4A/B; p=0.029 Bonferroni-corrected Kruskal–Wallis test, n = 6 ROIs). GABA+ cell density in L2/3 was significantly higher than that in all other layers except L4C (p=0.028 vs. L4A/B and p=0.013 vs. L6, p=0.007 vs. L1 and p=0.005 vs. L5). GABA density in L4C did not differ from any other layers, but the percent of GABA+ cells in L4C was significantly higher than that in L1 (p=0.009) and 4A/B (p=0.000022). As expected, within each layer, GABA+ cell density was significantly higher than PV+ cell density (p<0.05, one-sided *t*-test for equality of means, n = 6 ROIs).

We compared the laminar distributions of GABA+ and PV+ cell density in marmoset V1 with previously published distributions of these two cell markers in mouse V1 (*Xu et al., 2010*). In marmoset V1, there is an overall higher density of PV+ cells than in mouse V1, with density peaking in L2/3 and 4C. In contrast, PV+ cell density in mouse V1 peaks in L4 and 5, and density in L2/3 is lower than that in all other layers (*Figure 2C*). GABA+ cell density in marmoset V1 peaks in L2/3 followed by 4C, whereas it peaks in L4 and 5 with a smaller third peak in L2/3 in mouse V1 (*Figure 2D*).

Counts of cells double labeled for GABA+ and PV+ revealed that 92.3% ± 1.9 of PV+ cells across all layers were GABA+, and that PV+ cells represent on average 61.4% ± 2.7 of all GABA+ cells, ranging from 54.5% ± 3.7 in L6 to 78.5% ± 5.6 in L4C (*Figure 2E*). This differs from mouse V1 in which PV+ cells represent about 40% of all GABA cells (*Xu et al., 2010*).

## Laminar specificity and coverage of GABA-specific AAV-h56D

*Figure 3* shows tdT expression at three injection sites, each of a different serotype (9,7,1) of the GABA-specific AAV-h56D-tdT. Identical injection volumes of each serotype, delivered at three different cortical depths (see 'Materials and methods'), resulted in viral expression regions that differed in both size as well as laminar distribution, suggesting the different serotypes may have different capacity of infecting cortical neurons and layers. The AAV7 injection resulted in the smallest expression region, which additionally was biased to the superficial and deep layers, with only a few cells expressing tdT in the middle layers (*Figure 3B*). AAV9 (*Figure 3A*) and AAV1 (*Figure 3C*) resulted in larger expression regions, which involved all cortical layers. Given identical volumes and titers used for the AAV9 and AAV7 injections (injected volume of the AAV1 was the same but titer was higher; see *Supplementary*

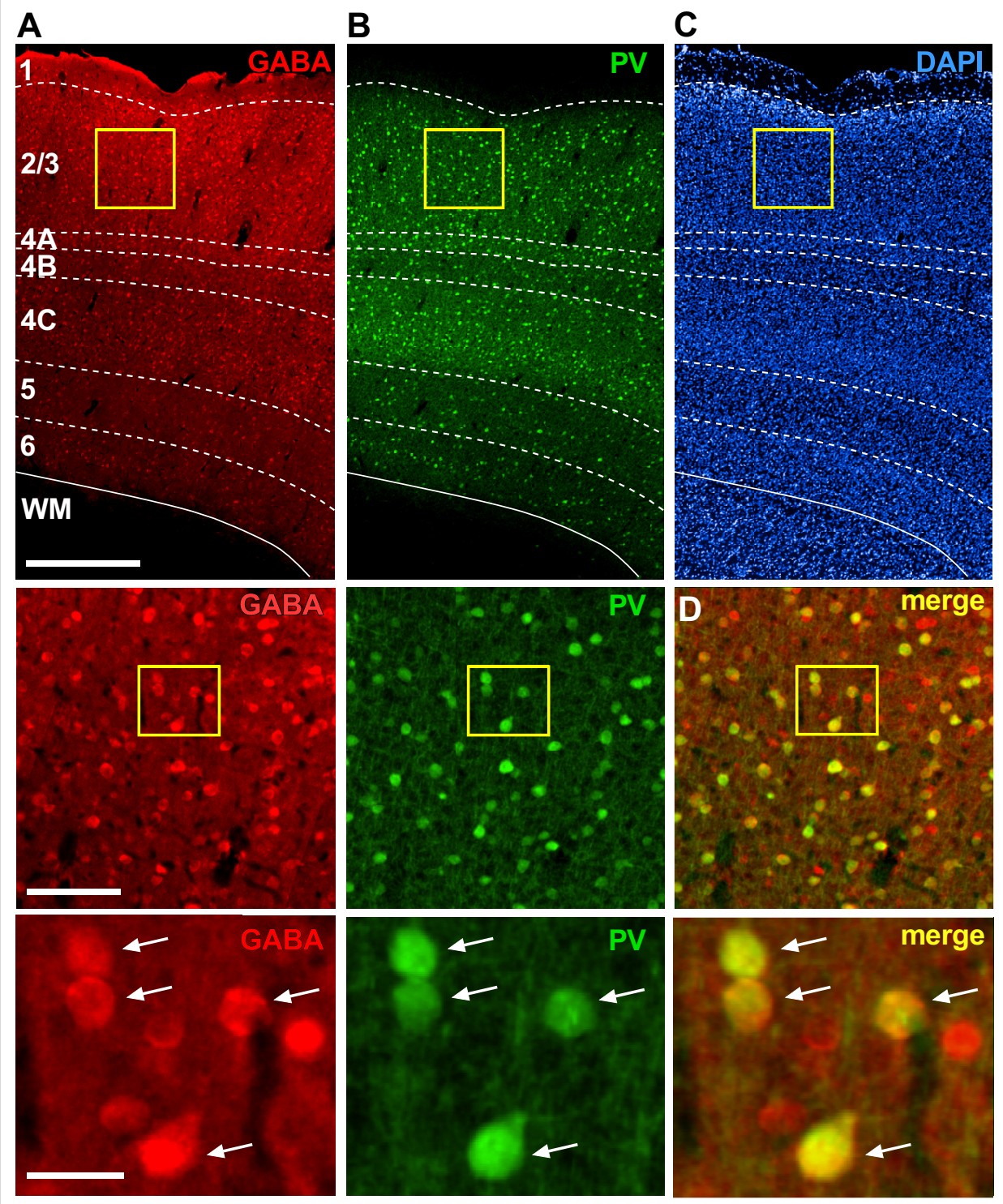

**Figure 1.** Laminar expression of GABA and PV immunoreactivity in marmoset V1. Epifluorescence images of the same V1 section triple-stained for GABA- (red channel) and PV- (green channel) immunohistochemistry (IHC) and DAPI (blue channel), showing individual and merged channels. (**A**) GABA+ expression through all cortical layers (top). *Dashed contours* mark layer boundaries; *solid contour* marks the bottom of the cortex. Cortical layers are indicated. Scale bar: 500 μm (valid for the top panels in **A–C**). Middle: V1 region inside the *yellow box* in (**A**) shown at higher magnification. The cells inside the *yellow box* are shown at higher magnification in the bottom panel. Scale bar: 100 μm (valid for the middle panels in **A, B** and the top panel in **D**). Bottom: scale bar: 25 μm (valid for the bottom panels in **A–C**). (**B**) Same as in (**A**) but for PV+ expression. (**C**) DAPI stain used to reveal cortical layers. (**D**) Merge of red (GABA) and green (PV) channels shown in the respective panels to the left. *Arrows* point to double-labeled cells.

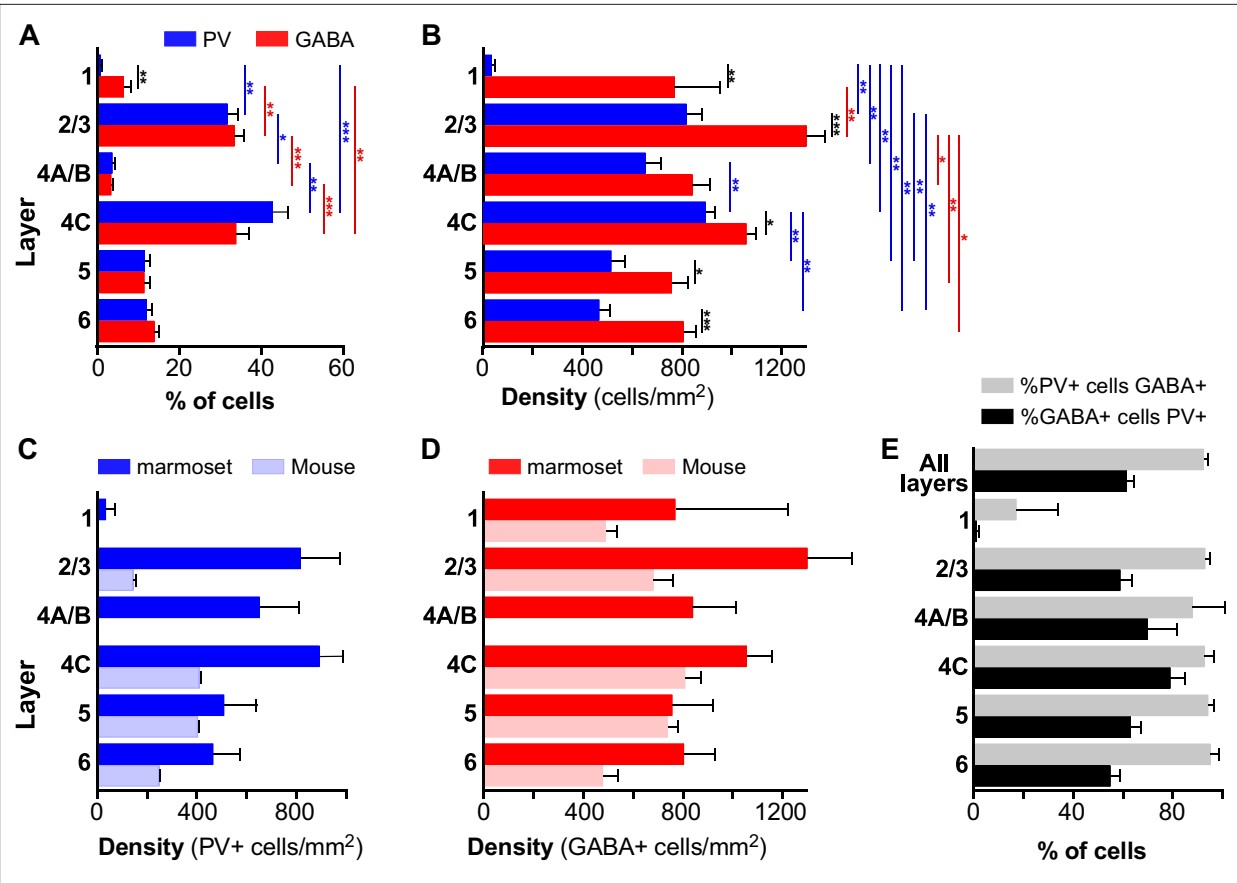

**Figure 2.** Laminar distribution of GABA+ and PV+ cells in marmoset V1. (**A**) Average percent of total number of GABA+ (*red*) or PV+ (*blue*) cells in each layer. Here and in (**B, E**) error bars represent standard error of the mean (s.e.m.) across regions of interest (ROIs) (n = 6 ROIs in **A, B, E**). In all panels *asterisks* indicate statistical significance (*<0.05, **<0.01, ***<0.001). (**B**) Mean density of GABA+ and PV+ cells in each layer. (**C**) Mean density of PV+ cells in marmoset (*dark blue*) and mouse (*light blue*) V1. Here and in (**D**), mouse data are from *Xu et al., 2010*, error bars represent the standard deviation, and n = 4–6 ROIs for mouse and 6 ROIs for marmoset. (**D**) Mean density of GABA+ cells in marmoset (*red*) and mouse (*pink*) V1. (**E**) Average percent of all counted PV+ cells that were double-labeled for GABA (*gray*), and average percent of all counted GABA+ cells that were double-labeled for PV+ (*black*) are shown at the top of the histogram. The percentages for each layer are shown underneath.

file 1), as well as identical post-injection survival times for all three serotypes, the differences in the size of the expression region are likely due to different tropism and/or viral spread of the different serotypes. However, given that we only made a single injection per serotype, we cannot exclude that other factors may have contributed to the reduced spread of the AAV7.

In *Figure 4*, we report quantitative counts for each serotype. Below we describe these results, but acknowledge that the differences we observed between serotypes need to be interpreted with caution, given they are based on a single injection per serotype. *Figure 4A* compares quantitatively tdT expression obtained with each viral serotype, quantified as the percent of total tdT+ cells in each layer for each serotype, with the percent laminar distribution of GABA+ cells identified by IHC. Serotypes 9 and 1 overall showed similar laminar distribution as GABA+ IHC, the percent of tdT+ cells peaking in L2/3 and 4C, suggesting good specificity of viral infection (the laminar distributions of AAV9- and AAV1-induced tdT+ cells were not significantly different from the GABA+ cell distribution; p>0.05 for all comparisons, Bonferroni-corrected independent-samples median test, n = 4–6 ROIs). In contrast, due to the lower capacity of AAV7 to infect the mid-layers, AAV7-induced tdT expression was relatively higher in L2/3 compared to GABA+ expression, approaching statistical significance (p=0.059; Bonferroni-corrected independent-samples median test, n = 4 AAV7 ROIs and 6 GABA-IHC ROIs). The percent of AAV7-infected cells was also significantly higher than the percent of AAV9- and/ or AAV1-infected cells in L2/3 (p=0.028 for both comparisons) and in L6 (p=0.028 for AAV7 vs. AAV1),

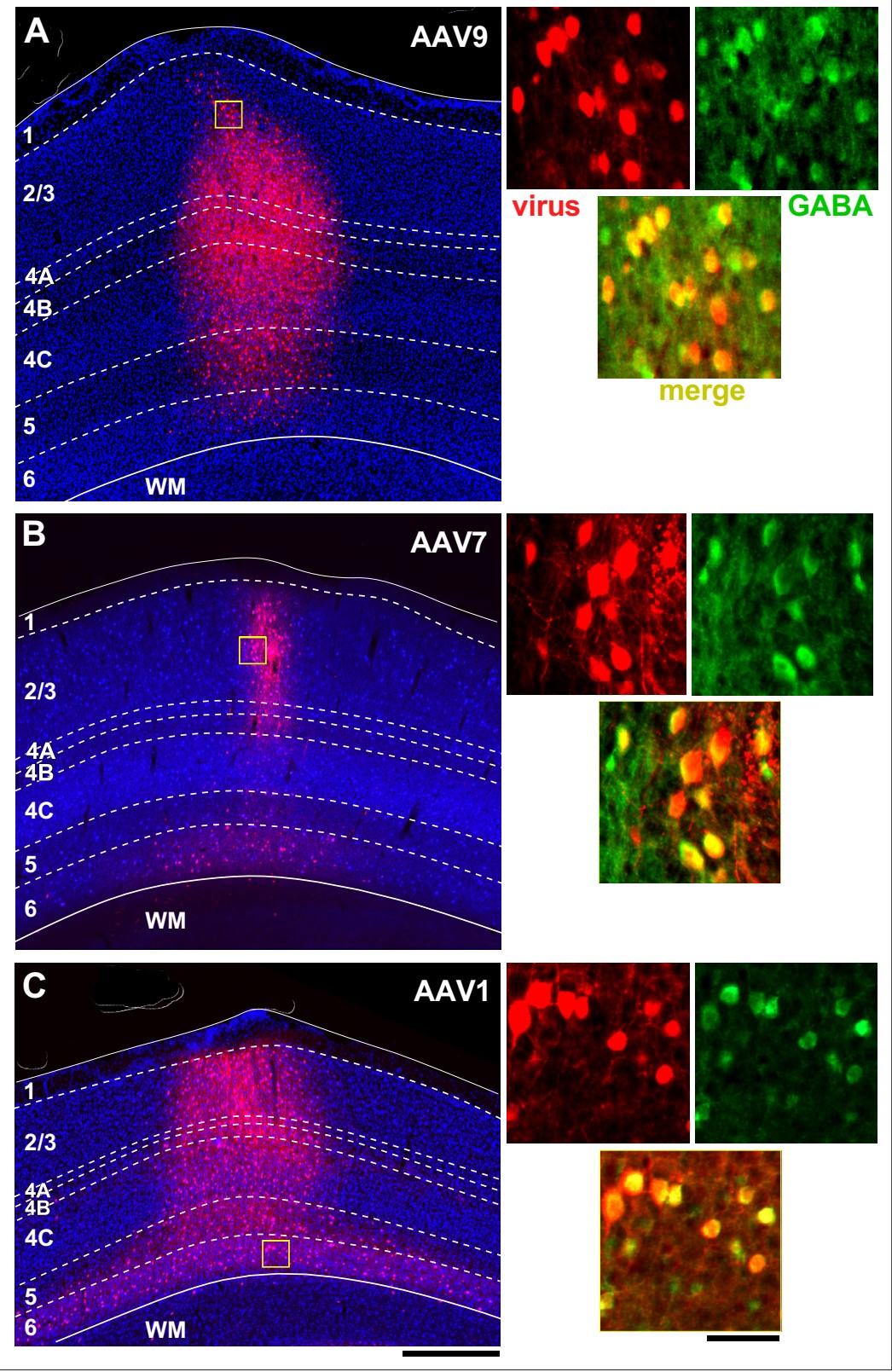

**Figure 3.** Laminar profile of pAAV-h56D-mediated tdT expression in marmoset V1. (**A–C**) Left: tdT expression (*red*) across V1 layers (indicated) following injection of an identical volume of AAV-h56D-tdT serotype 9 (**A**), serotype 7 (**B**), and serotype 1 (**C**). The viral titers for the AAV9 and AAV7 injections were also the same, while titer was higher for AAV1 (see *Supplementary file 1*). The tdT expression region in panel (**A**) is a merge of two adjacent sections

*Figure 3 continued on next page*

*Figure 3 continued*

because the tdT expression region did not encompass all layers in individual sections. TdT expression in other panels, instead, is from a single section. *Dashed contours* mark layer boundaries; *solid contours* mark the top and bottom of the cortex. Layers were identified based on DAPI counterstain (*blue*). Note that the cortical thickness varies across cases because these sections are from different regions of V1. *Yellow box* in each panel is the region shown at higher magnification on the right. Scale bar: 500 μm (valid for **A–C**). Right: Higher magnification of the V1 region inside the *yellow box* in each respective left panel, showing individual channels (*red*: viral-mediated tdT expression; *green*: GABA + IHC) and the merge of these two channels (*yellow*). Scale bar: 50 μm (valid for **A–C**).

and significantly lower than the percent of AAV9- and AAV1-infected cells in L4C (p=0.028 for both comparisons; Bonferroni-corrected independent-samples median test, n = 4 ROIs).

To quantify the specificity of tdT expression induced by each serotype, that is, the accuracy in inducing tdT expression selectively in GABA cells, for each serotype separately we measured the percent of tdT-expressing cells that colocalized with GABA expression revealed by IHC (*Figure 4B*). Overall, across all layers, AAV9 showed the highest specificity (82.3% ± 1.1) followed by AAV7 (79.2% ± 5.4), and AAV1 (75.3% ± 2.6), and there was a statistically significant difference in overall specificity between AAV9 and AAV1 (p=0.014; Bonferroni-corrected independent-samples median test, n = 4 ROIs for each serotype). The specificity of AAV9 did not differ significantly across layers, ranging from 80.4% ± 2.1 in L4C to 93.8% ± 6.3 in L4A/B. In contrast, specificity for the other two serotypes varied by layer. AAV7 showed highest specificity in L1 (100% but there were only two tdT+ cells in this layer), L4C (90.1% ± 5.9) and L6 (87.1% ± 8.1), and lowest in L4A/B (50% ± 35.4). AAV1 specificity was highest in L6 (85.4% ± 8.8) and L4C (81.6% ± 1.4) and lowest in L4A/B (38.3% ± 21.7). There was a tendency for AAV9 to be more specific than AAV1 in L4A/B (93.8% ± 6.3 vs. 38.3% ± 21.7) and L5 (88.8% ± 6.6 vs. 59.4% ± 8.3) but these differences did not reach statistical significance.

To quantify the efficiency of the virus in inducing tdT expression in GABA cells, for each serotype separately we measured the viral coverage as the percent of GABA+ cells within the viral injection site that colocalized with tdT expression (*Figure 4C*). Overall, across all layers, AAV9 and AAV1 showed significantly higher coverage (66.1% ± 3.9 and 64.9% ± 3.7) than AAV7, which showed much lower coverage values (34% ± 5.6; p=0.014 for both comparisons; Bonferroni-corrected independent-samples

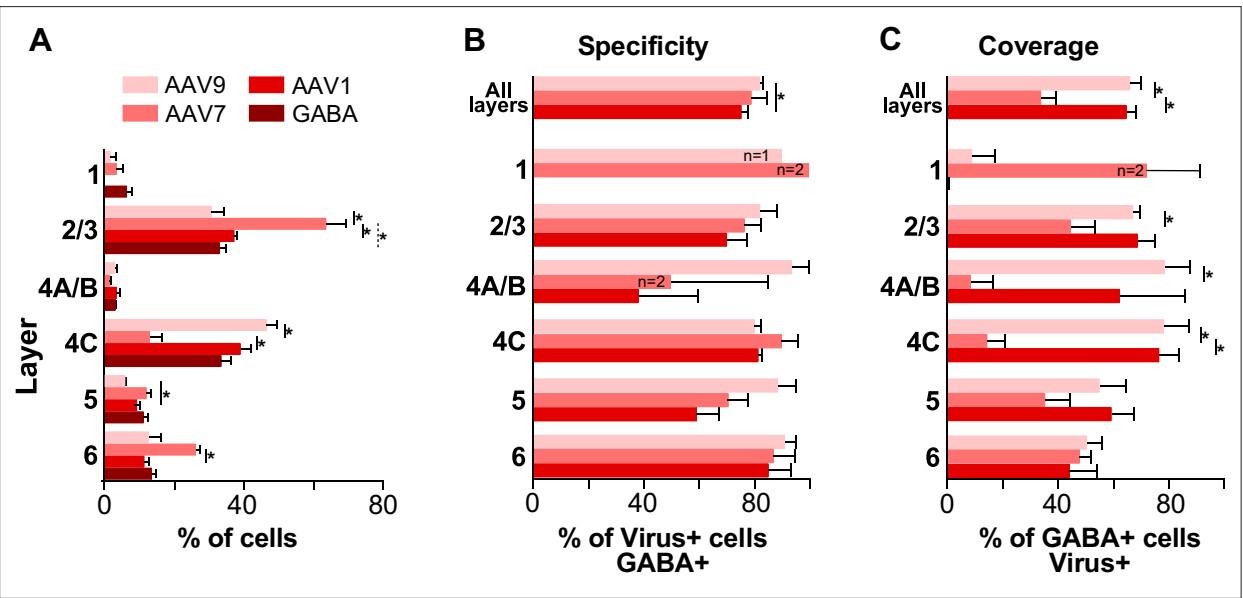

**Figure 4.** Laminar distribution, specificity, and coverage of tdT expression induced by three different serotypes of pAAV-h56D. (**A**) Average percent of total number of GABA immunoreactive cells, and average percent of total number of tdT-expressing cells after injections of three different serotypes of the GABA-AAV vector, in each V1 layer. (**B**) Specificity of tdT expression induced by each serotype across all layers and in each layer. Specificity is defined as the percent of viral-mediated tdT expressing cells that colocalize with GABA immunoreactivity. (**C**) Coverage of each viral serotype across all layers and in each layer, defined as percent of GABA immunoreactive cells that co-express tdT. In all panels, error bars represent s.e.m. across regions of interest (ROIs) (n = 4 for AAV9, 4 for AAV7, 4 for AAV1, 6 for GABA-IHC), and *asterisks* indicate statistically significant differences at the p<0.05 level.

median test, n = 4 ROIs across two sections for each serotype). AAV9 and AAV1 coverage was similar across layers, and both showed slightly higher coverage in superficial (AAV9: 67.4% ± 2.5; AAV1: 69% ± 6.5) and middle layers (AAV9: 78.5% ± 9.1; AAV1: 76.9% ± 7.4), compared to deep layers (AAV9: 50–55%, AAV1: 44–60%). Instead, AAV7 showed very low coverage values in L4A/B (8.3% ± 8.3) and L4C (14.4% ± 6.7) and highest values in L6 (47.9% ± 4.3) followed by L2/3 (44.6% ± 9). AAV7 coverage was significantly lower than AAV9 coverage in L2/3 (p=0.014), L4A/B (p=0.014), and L4C (p=0.014), and was significantly lower than AAV1 in L4C (p=0.014; Bonferroni-corrected independent-samples median test, n = 4 ROIs for each serotype). Thus, our results suggest that AAV9 is the serotype of choice for marmoset studies of GABAergic neurons requiring highest specificity and coverage across all layers, but AAV7 may be a better choice for studies intending to restrict transgene expression to L6 or L2/3 GABA cells with good specificity.

## Laminar specificity and coverage of PV-specific AAV-PHP.eB-S5E2

We assessed the laminar specificity and coverage of the AAV-PHP.eB-S5E2-tdT following injections of different viral volumes ranging from 90 nl to 585 nl (see *Supplementary file 1*). *Figure 5* shows fluorescent images of tdT expression at the site of viral injection for an example 105 nl injection (*Figure 5A*) and an example 315 nl injection (*Figure 5B*).

*Figure 6A* compares tdT expression resulting from injections of different volumes, quantified as the percent of total tdT+ cells in each layer for each volume, with the percent laminar distribution of PV+ cells identified by IHC. Cell counts from injections of 315–585 nl volumes were pooled as injections ≥ 315 nl produced similar results (see *Supplementary file 2*). We found that the distribution of tdT expression resulting from all injection volumes did not differ significantly from the distribution of PV+ IHC, all distributions similarly peaking in L2/3 and 4C (p>0.1 for all comparisons; Bonferroni-corrected Kruskal–Wallis test; n = 8–12 PV-AAV ROIs, and 6 PV-IHC ROIs), suggesting good viral specificity.

The specificity of tdT expression induced by different injection volumes is quantified in *Figure 6B*, separately for three groups of injection volumes: group 1 = 90–105 nl, group 2 = 180 nl, group 3 = 315–585 nl. Results from individual injection cases are reported in *Supplementary file 2*. The degree of viral specificity was high at all volumes, but depended slightly on injection volume. Overall, across all layers, group 2 (180 nl) showed the highest specificity (94.7% ± 1.6), which differed significantly from the specificity of group 3 volumes (≥315 nl; 82% ± 3.2; p=0.01, Bonferroni-corrected Kruskal–Wallis test, n = 8–12 ROIs). Specificity was similar across layers for all volumes, but volumes ≥ 315 nl resulted in slightly lower specificity than smaller volumes in L4C (76.6% ± 5.6 for >315 nl volumes vs. 95.2% ± 1.7 and 94.9% ± 3 for 90–105 nl and 180 nl volumes, respectively) and L5 (80.1% ± 5.8 for ≥315 nl vs. 97.9% ± 2.1 and 97% ± 1.9 for 90–105 and 180 nl, respectively), and these differences in L5 were statistically significant (p=0.013 and 0.005; Bonferroni-corrected Kruskal–Wallis test; n = 8–12 ROIs).

The viral coverage resulting from each injection volume is quantified in *Figure 6C* separately for the three different volume groups and shown for each individual injection case in *Supplementary file 2*. Coverage of the AAV-PHP.eB-S5E2-tdT was high, did not depend on injection volume, and it was similar across layers for all volumes. Overall, across all layers coverage ranged from 78% ± 1.9 to 81.6% ± 1.8.

## Reduced GABA and PV immunoreactivity at the viral injection site

Qualitative observations of tissue sections seemed to indicate slightly reduced expression of both GABA and PV immunoreactivity at the viral injection sites, extending beyond the borders of the injection core (*Figure 7—figure supplement 1*). To quantify this observation, we counted GABA+ and PV+ cells at the site of the viral injections (n = 12 ROIs across 6 sections for AAV-h56D injection sites [pooled across serotypes], and 28 ROIs across 14 sections for AAV-S5E2 injection sites) and at sites located several millimeters beyond the viral injection borders (n = 6 ROIs across three sections). We found that both the number and density of GABA+ and PV+ cells were reduced across all layers at the site of the AAV-h56D (*Figure 7A–D*) and AAV-S5E2 (*Figure 7E–H*) injections compared to control tissue away from the injection sites. The magnitude of the reduction in immunoreactivity depended on the viral type, with the AAV-h56D virus inducing an overall greater and more significant reduction in GABA immunoreactivity (28.1% and 21.5% reduction in mean GABA+ cell number and density across all layers, respectively, p=0.024 in *Figure 7A* and p=0.013 in *Figure 7B*, Mann–Whitney *U* test) than in PV immunoreactivity (20.5 and 10.2% reduction in mean PV cell number and density across

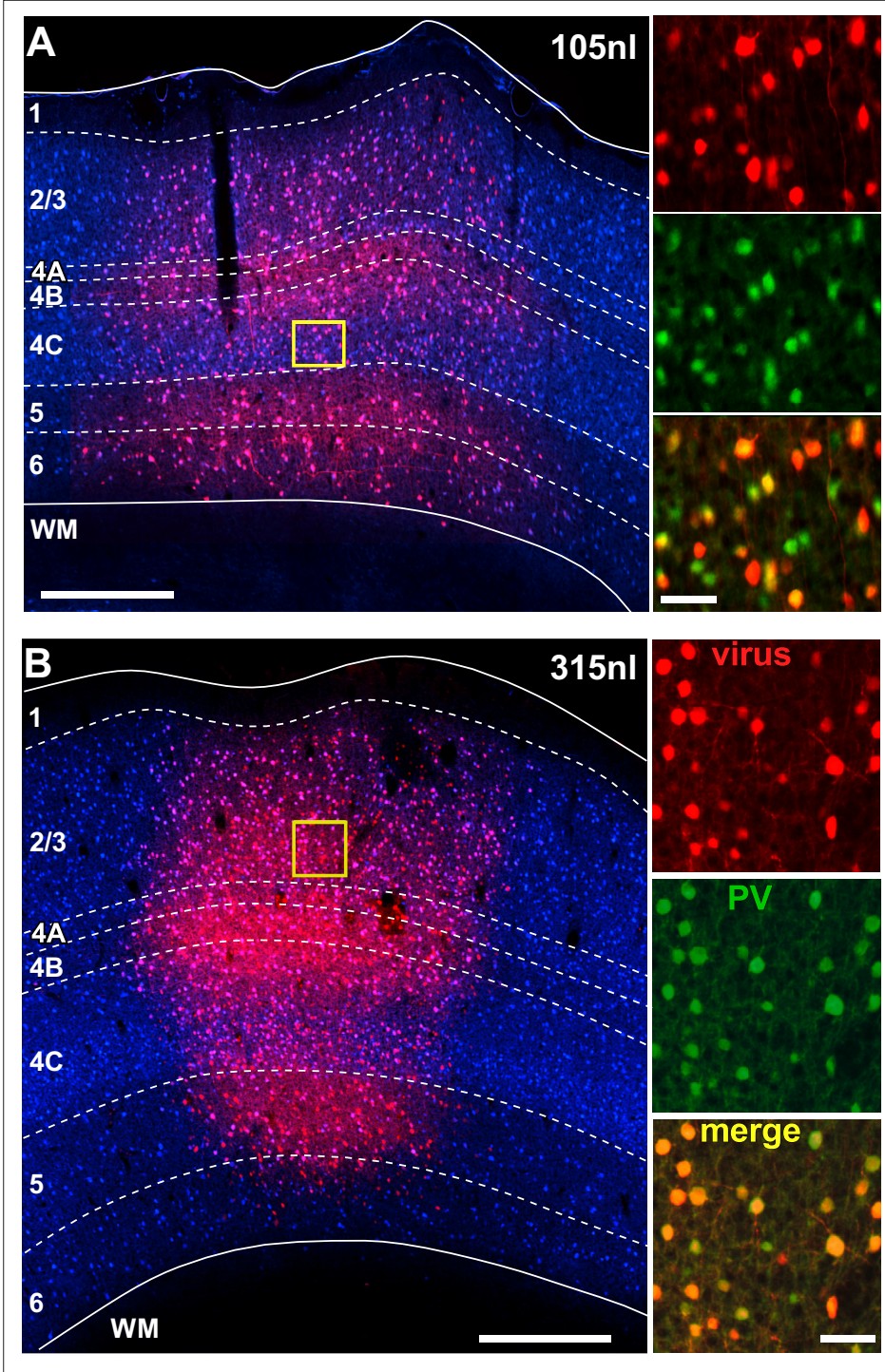

**Figure 5.** Laminar profile of AAV-PHP.eB-S5E2-mediated tdT expression in marmoset V1. (**A**) Left: tdT expression (*red*) across V1 layers following an injection of 105 nl volume of AAV-PHP.eB-S5E2-tdT. *Dashed contours* mark layer boundaries; *solid contours* mark the top and bottom of the cortex. Layers were identified based on DAPI counterstain (*blue*). *Yellow box* is the region shown at higher magnification in the right panels. Scale bar here and in the left panel in (**B**): 500 μm. Right: higher magnification of the V1 region inside the *yellow box* in the left panel, showing individual channels (*red:* viral-mediated tdT expression; *green:* PV+ IHC) and the merge of these two channels (*yellow*). Scale bar here and in the right panels in (**B**): 50 μm. (**B**) Same as in (**A**) but for an injection volume of 315 nl.

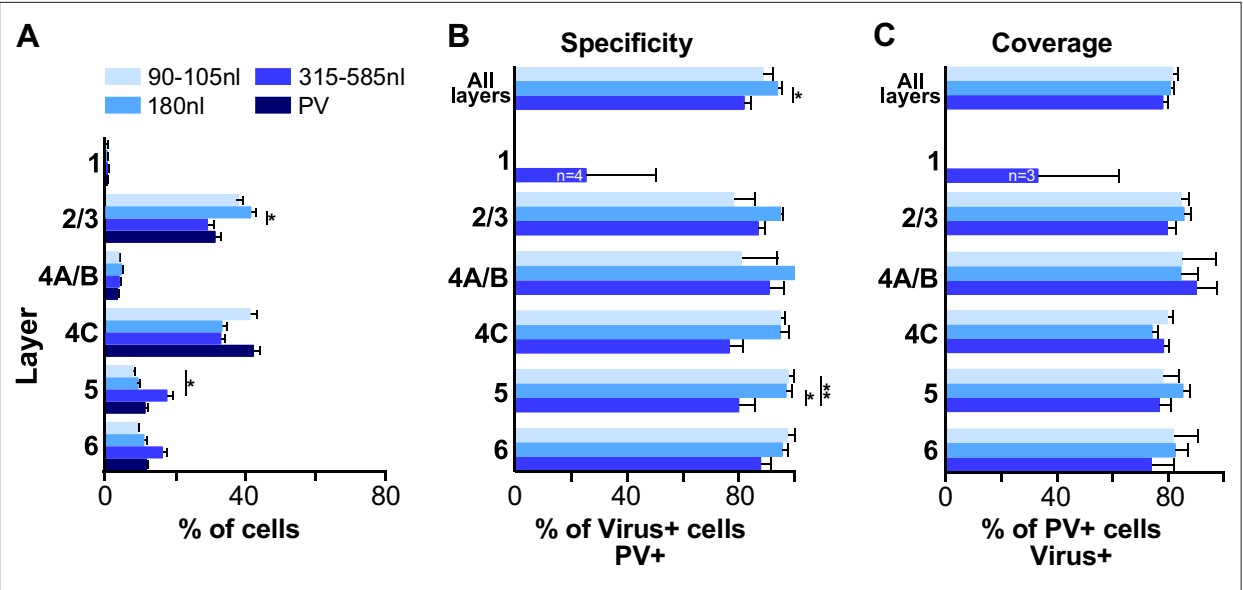

**Figure 6.** Laminar distribution, specificity, and coverage of tdT expression induced by three different injection volumes of AAV-PHP.eB-S5E2. (**A**) Average percent of total number of PV immunoreactive cells, and average percent of total number of tdT-expressing cells after injections of 3 different volumes of the PV-AAV vector, in each V1 layer. (**B**) Specificity of tdT expression induced by each injection volume across all layers and in each layer. (**C**) Coverage of each viral injection volume across all layers and in each layer. In all panels, error bars represent s.e.m. across regions of interest (ROIs) (n = 8 for 90–105 nl, 8 for 180 nl, 12 for 315–585 nl PV-AAV injection volumes and 6 for PV-IHC), and *asterisks* indicate statistically significant differences.

all layers, respectively; p=0.041 in *Figure 7C* and p=0.125 in *Figure 7D*, Mann–Whitney *U* test) and vice versa for the AAV-S5E2 virus, which reduced PV immunoreactivity (33.3% and 25.4% reduction in mean PV+ cell number and density across all layers, respectively, p<0.001 in *Figure 7G* and p=0.013 in *Figure 7H*) more than GABA immunoreactivity (27.4% and 20.2% reduction in mean GABA+ cell number and density across all layers, respectively, p=0.005 in *Figure 7E* and p=0.042 in *Figure 7F*). The reduced GABA and PV immunoreactivity caused by the viruses imply that the specificity of the viruses we have validated in this study is likely higher than estimated in *Figures 4 and 6*. Moreover, reduced GABA and PV immunoreactivity could at least partly underlie the apparent reduction in specificity observed for larger PV-AAV injection volumes (see 'Discussion').

## Discussion

Understanding the connectivity and function of inhibitory neurons and their subtypes in primate cortex requires the development of viral tools that allow for specific and robust transgene expression in these cell types. Recently, several enhancer and promoter elements have been identified that allow to selectively and efficiently restrict gene expression from AAVs to GABAergic neurons and their subtypes across several species, but a thorough validation and characterization of these enhancer-AAVs in primate cortex is lacking. In particular, previous studies have not characterized the specificity and coverage of these vectors across cortical layers. In this study, we have characterized two main enhancer-AAV vectors designed to restrict expression to GABAergic cells or their PV subtypes, which show high specificity and coverage in marmoset V1. Specifically, we have shown that the GABA-specific AAV9-h56D (*Mehta et al., 2019*) induces transgene expression in GABAergic cells with up to 91–94% specificity and 80% coverage, depending on layer, and the PV-specific AAV-PHP.eB-S5E2 (*Vormstein-Schneider et al., 2020*) induces transgene expression in PV cells with up to 98% specificity and 86–90% coverage, also depending on layer. We conclude that these two viral vector types provide useful tools to study inhibitory neuron connectivity and function in primate cortex.

Many recent studies have investigated the connectivity and function of distinct classes of inhibitory neurons in mouse V1 and other cortical areas (*Tremblay et al., 2016*; *Wood et al., 2017*; *Shin and Adesnik, 2023*). In contrast, similar studies in the primate have been missing due to the lack of tools to selectively express transgenes in specific cell types and the difficulty of performing genetic

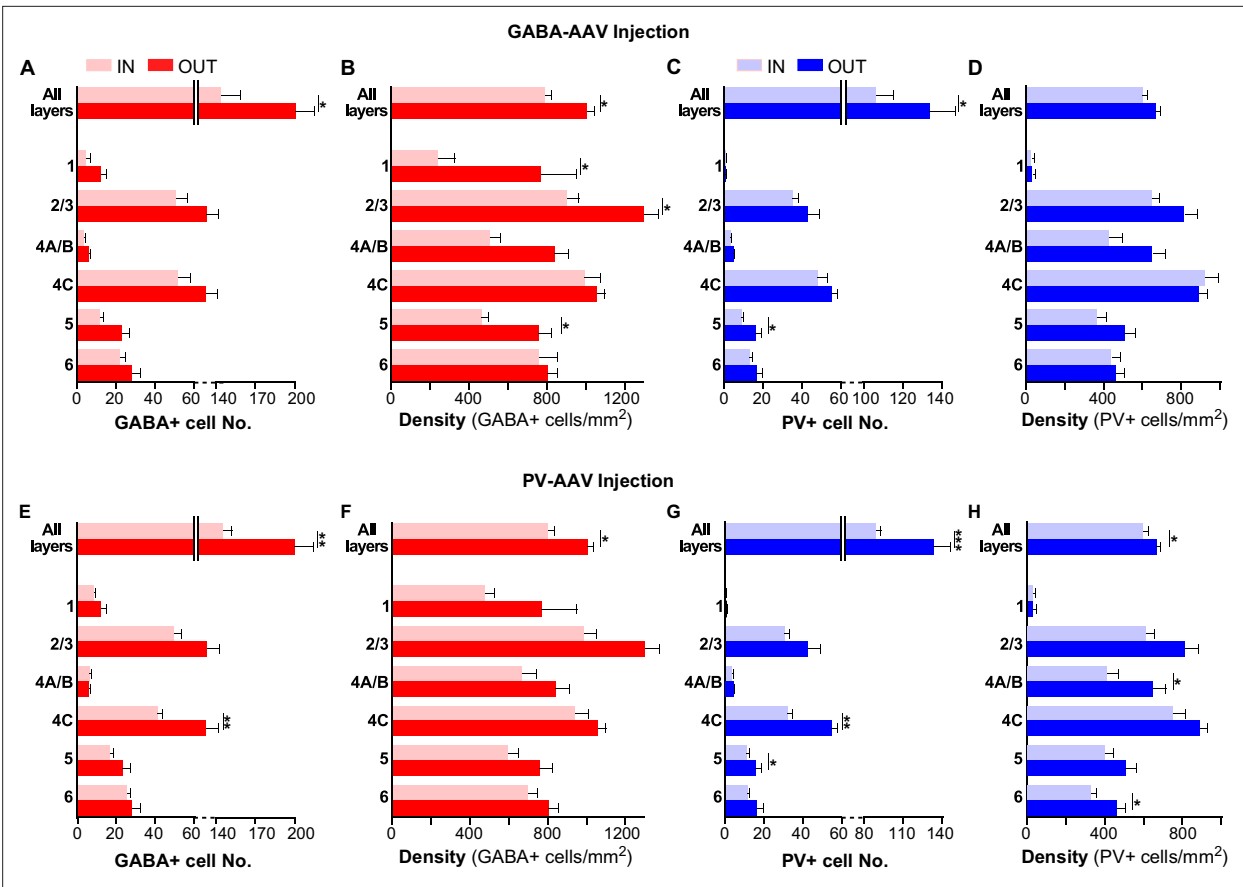

**Figure 7.** Reduced GABA and PV immunoreactivity at the viral injection site. (**A, B**) Number (**A**) and density (**B**) of GABA+ cells inside (*pink*; n = 12 regions of interest [ROIs] across six sections) and outside (*red*; n = 6 ROIs across three sections) the GABA-AAV injection sites. (**C, D**) Number (**C**) and density (**D**) of PV+ cells inside (*light blue*; n = 12 ROIs across six sections) and outside (*dark blue*; n = 6 ROIs across three sections) the GABA-AAV injection sites. (**E, F**) Number (**E**) and density (**F**) of GABA+ cells inside (*pink*; n = 28 ROIs across 14 sections) and outside (*red*; n = 6 ROIs across three sections) the PV-AAV injection sites. (**G, H**) Number (**G**) and density (**H**) of PV+ cells inside (*light blue*; n = 28 ROIs across 14 sections) and outside (*dark blue;* n = 6 ROIs across three sections) the PV-AAV injection sites. Error bars: s.e.m. *Asterisks:* statistically significant comparisons. In each panel, statistical comparisons across layers were performed using the Bonferroni-corrected Kruskal–Wallis or independent-samples median tests; comparisons between total IN and OUT populations in each panel were performed using the Mann–Whitney *U* test.

The online version of this article includes the following figure supplement(s) for figure 7:

**Figure supplement 1.** Reduced GABA and PV immunoreactivity at the viral injection site.

manipulation in this species. It is important to study inhibitory neuron function in the primate because it is unclear whether findings in mice apply to higher species, and inhibitory neuron dysfunction in humans has been implicated in several neurological and psychiatric disorders (**Marín, 2012**; **Goldberg and Coulter, 2013**; **Lewis, 2014**). While the basic inhibitory neuron subtypes seem to exist across most mammalian species studied (**DeFelipe, 2002**), species differences may exist, particularly given the evolutionary distance between mouse and primate. Indeed, species differences have been reported in marker expression patterns (**Hof et al., 1999**), in the proportion of cortical GABAergic neurons (24–30% in primates vs. 15% in rodents) (**Hendry et al., 1987**; **Beaulieu, 1993**), in the proportion of PV neurons (74% in macaque V1 vs. 30–40% in mouse) (**Van Brederode et al., 1990**; **Defelipe et al., 1999**; **Xu et al., 2010**), and in the abundance of the various subtypes (**Krienen et al., 2020**). Here, we found that PV cells in marmoset V1 across all layers represent on average 61% of all GABAergic cells, and up to 79% in V1 L4C. These percentages are lower than previously reported for macaque V1 by **Van Brederode et al., 1990** (74% across all layers and nearly 100% in L4C), but higher than recently reported by **Kelly et al., 2019** (52% across all V1 layers, up to 80% in L4C). We also found differences in the V1 laminar expression of both GABA+ and PV+ cells between mouse and marmoset. Specifically, GABA+ and PV+ expression peaks in L2/3 and 4C in marmoset V1, but in

L4 and 5 in mouse V1. Similar differences between mouse and primate V1 in the laminar distribution of PV cells were reported previously (*Kooijmans et al., 2020*; *Medalla et al., 2023*). Our results on the laminar distribution of PV and GABA immunoreactivity are consistent with previous qualitative and quantitative studies in macaque V1 (*Hendry et al., 1989*; *Blümcke et al., 1990*; *Defelipe et al., 1999*; *Disney and Aoki, 2008*; *Kelly et al., 2019*; *Kooijmans et al., 2020*; *Medalla et al., 2023*), and with a quantitative study in marmoset V1 (*Goodchild and Martin, 1998*).

We compared laminar distribution, specificity, and coverage of three different serotypes of the AAV-h56D vector. Serotypes 9 and 7 showed slightly greater specificity than serotype 1, and the specificity of AAV9 was more consistent across layers than the specificity of serotypes 7 and 1, which instead varied somewhat across layers. Serotypes 9 and 1 showed greater coverage than serotype 7. Thus, serotype 9 may be a better choice when high specificity and coverage across all layers are required. Serotype 7, instead, showed high specificity (80%) but low coverage (34%), except in layer 6 (48%) and L2/3 (45%); therefore, this serotype may be desirable to restrict transgene expression to L6 or 2/3 GABAergic cells. We note that these differences among serotypes should be interpreted with caution as they are based on a single injection per serotype. Despite this, our results demonstrate sufficiently high efficiency and specificity of transgene expression in GABA cells using the h56D promoter, at least with two of the three AAV serotypes we tested, warranting their use in the non-human primate.

We compared laminar distribution, specificity, and coverage resulting from different volume injections of the AAV-PHP.eB-S5E2 vector. Injections of 180 nl volume resulted in higher specificity (95% across layers) and coverage (81% across all layers) than obtained with injection volumes equal to or larger than 315 nl (specificity 82% and coverage 78% across all layers), although coverage did not differ significantly across volumes. This mild dose-dependent alteration of specificity could depend on some off-target expression reaching above detection levels at higher doses. Thus, injection volumes of 150–300 nl are recommended for studying PV neuron function and connectivity using this viral vector. Alternatively, or in addition, an apparent dose-dependent reduction in specificity may result from viral-induced suppression of PV immunoreactivity, which could be more pronounced for larger volume injections. Indeed, we found that both GABA- and PV-specific AAVs slightly, but significantly, reduced both GABA and PV immunoreactivity at the site of the viral injection, but GABA expression was more reduced at the AAV-h56D injection site, while reduction in PV expression was more marked at the AAV-S5E2 injection site. This reduction in GABA and PV immunoreactivity at the viral injected sites most likely affected our measurements of specificity, suggesting that specificity for the viruses tested in this study is even higher than revealed by our counts. However, this reduced immunoreactivity raises concerns about the virus or the high level of reporter protein possibly harming the cell physiology. Our data does not allow us to assess the origin of the reduced GABA and PV immunoreactivity. Qualitative observations did not reveal structural damage at the site of the viral injections to suggest cell death. Moreover, we have been able to record the electrophysiological responses of V1 neurons in which opsin protein expression was induced via injections of these viruses (*Vafaei et al., 2024*). Notably, viral-induced downregulation of gene expression in host cells, including of inhibitory neuron marker genes such as *PV*, has been documented for other viruses, such as rabies virus (*Prosniak et al., 2001*; *Zhao et al., 2011*; *Patiño et al., 2022*). As such, it is possible that subtle alteration of the cortical circuit upon parenchymal injection of viruses (including AAVs) leads to alteration of activity-dependent expression of *PV* and *GABA*.

## Materials and methods

### Experimental design

Enhancer-AAV vectors carrying the gene for the reporter protein tdTomato (tdT) were injected in area V1 of marmoset monkeys. After an appropriate survival time, the animals were euthanized. The brains were processed for histology and IHC to identify GABA+ and PV+ cells and cortical layers. The laminar distribution of GABA+ and PV+ cells, and of viral-mediated tdT expression was analyzed quantitatively.

### Animals

Four female common marmosets (*Callithrix jacchus*) between the ages of 2 and 8 years old (weight about 500 g), obtained from the University of Utah in-house colony, were used in this study. All

procedures were approved by the University of Utah Institutional Animal Care and Use Committee (IAUC protocol no. 21-12015) and conformed to the ethical guidelines set forth by the USDA and NIH.

## Surgical procedures

Animals were pre-anesthetized with alfaxalone (10 mg/kg, i.m.) and midazolam (0.1 mg/kg, i.m.) and an IV catheter was placed in either the saphenous or tail vein. To maintain proper hydration Lactated Ringers solution was continuously infused at 2–4 cc/kg/hr. The animal was then intubated with an endotracheal tube, placed in a stereotaxic apparatus, and artificially ventilated. Anesthesia was maintained with isoflurane (0.5–2.5%) in 100% oxygen. Throughout the experiment, end-tidal $CO_2$, ECG, blood oxygenation, and rectal temperature were monitored continuously.

Under aseptic conditions, the scalp was incised and several small (~2 mm) craniotomies and durotomies were made over dorsal V1. A single injection of a viral vector was made into each craniotomy. On completion of the injections, each craniotomy was filled with Gelfoam and sealed with dental cement, the skin was sutured, and the animal was recovered from anesthesia. Animals survived 3–4 weeks (one animal survived 2 weeks) post-injections (*Supplementary file 1*), to allow for viral expression, and were sacrificed with beuthanasia (0.22 ml/kg, i.p.) and perfused transcardially with saline for 2–3 min, followed by 4% paraformaldehyde in 0.1 M phosphate buffer for 15–20 min.

## Injection of viral vectors

A total of 10 viral injections were made in four marmosets (*Supplementary file 1*). Each of two animals received one injection, and one animal five injections (three in one hemisphere and two in the other hemisphere) of AAV-PHP.eB-S5E2.tdTomato, obtained from the Dimidschstein laboratory (*Vormstein-Schneider et al., 2020*). The fourth animal received three injections, each of a different AAV serotype (1, 7, and 9) of the AAV-h56D-tdTomato (*Mehta et al., 2019*), obtained from the Zemelman laboratory (UT Austin). Viral vectors were loaded in glass micropipettes (tip diameter 30–45 μm) and pressure injected using a PicoPump (World Precision Instruments). To ensure viral infection of all cortical layers, each injection was made at three depths within the cortical column: 1.2–1.5 mm from the cortical surface (deep), 0.8–1.0 mm (middle), and 0.4–0.6 mm (superficial). After injecting at each depth, the pipette was left in place for 2–4 min before being retracted to the next depth, and for 5 min before being fully retracted from the brain. The PV-specific AAV was injected at four different total volumes: 585 nl (one injection), 315 nl (two injections), 180 nl (two injections), and 90–105 nl (two injections). The AAV-h56D vectors were each injected at a total volume of 600 nl. One-third of each total volume per injection was slowly (6–15 nl/min) injected at each of the three depths. For animals that received multiple injections in the same hemisphere, injections were spaced at least 3 mm apart to ensure no overlap. Viral titers and volumes of each injection as well as post-injection survival times for each case are reported in *Supplementary file 1*.

## Viral preparation

### AAV-PHP.eB-S5E2.tdT

Details about AAV- PHP.eB-S5E2.tdTomato cloning and production are provided in the original publication (*Vormstein-Schneider et al., 2020*). Briefly, the E2 enhancer sequence was amplified from mouse genomic DNA using the primer aatctaacatggctgctata and caattgctcagagttatttt (618 bp). Enhancer, reporter, and effector cloning was performed using the Gibson Cloning Assembly Kit (New England BioLabs, Cat# NEB-E5510S) following standard procedures. Specifically, for AAV-E2-SYP-dTomato, we amplified the SYP–tdTomato coding sequence from the plasmid Addgene no. 34881. The rAAVs were produced using standard production methods. Polyethylenimine was used for transfection and OptiPrep gradient (Sigma) was used for viral particle purification. Titer was estimated by quantitative PCR with primers for the WPRE sequence. The batch used in this study had a titer of 8.3 × $10^{12}$ viral genomes/ml.

### AAV-h56D.tdT

Details about AAV-h56D.tdTomato cloning and production are provided in the original publication (*Mehta et al., 2019*). Briefly, viruses were assembled using a modified helper-free system (Stratagene) as the indicated serotypes (rep/cap). Viruses were purified on sequential cesium gradients

according to published methods (*Grieger et al., 2006*). Titers were measured using a payload-independent qPCR technique (*Aurnhammer et al., 2012*). Typical titers were $1 \times 10^{13}$ - $1 \times 10^{14}$ viral genomes/ml.

## Histology and Immunohistochemistry

Area V1 was dissected away from the rest of the visual cortex. The block was postfixed for 3–12 hr in 4% paraformaldehyde, sunk in 30% sucrose for cryoprotection, and frozen sectioned in the parasagittal plane at 40 μm thickness. In one case (MM423, which received a 315 nl injection of AAV-PHP.eB-S5E2.tdTomato), the brain was sunk in a 20% glycerol solution and frozen at –80°C for 6 months prior to being sectioned. To locate the viral injection sites, a 1:5 series of tissue sections were wet-mounted and observed under microscopic fluorescent illumination. Sections containing each injection site had their coverslips removed, and fluorescent IHC was performed on free-floating sections to reveal both GABA+ and PV+ neurons. No IHC was performed to enhance reporter proteins signals as these were sufficiently bright. GABA and PV IHC was performed by incubating sections for 3 days at 4°C in primary antibody, followed by 12 hr incubation at room temperature in secondary antibody. The primary and secondary antibodies used for GABA-IHC were a rabbit anti-GABA antibody (1:200; Sigma-Aldrich, Burlington, MA; RRID:AB_477652) and an Alexa Fluor 647 AffiniPure Donkey Anti-Rabbit IgG (H+L) (1:200; Jackson ImmunoResearch Laboratory Inc, West Grove, PA; RRID:AB_2492288), respectively. The primary and secondary antibodies used for PV-IHC were a guinea pig anti-parvalbumin antibody (1:1000; Swant, Burgdorf, Switzerland; RRID:AB_2665495) and an Alexa Fluor 488 AffiniPure Donkey Anti-Guinea Pig IgG (H+L) (1:200; Jackson ImmunoResearch Laboratories Inc; RRID:AB_2340472), respectively. The sections were then mounted and coverslipped with Vectashield Antifade Mounting Medium with DAPI (Vector Laboratories, Newark, CA).

## Data analysis

Multi-channel wide-field fluorescent images of V1 tissue sections containing an injection site spanning all layers were acquired at 5–7 depths in the z plane using a Zeiss AxioImager Z2 fluorescent microscope equipped with a ×10 objective. Images were stitched, rotated, and cropped as necessary using Zen Blue software (Carl Zeiss AG) and loaded into Neurolucida software (MBF Bioscience) for data quantification. To quantify inhibitory neurons that expressed the viral-mediated reporter protein tdTomato, GABA+ and PV+ neurons revealed by IHC at the viral injection sites (i.e., the data shown in *Figures 4 and 6*, and the 'IN' data in *Figure 7*), we counted single-, double-, and triple-labeled cells across two 100-μm-wide ROIs extending through all layers at the injection site on each channel, yielding a total of four ROIs across two tissue sections being counted and analyzed for each viral injection site. All ROIs used for counts were positioned at the center of the viral expression region in sections where the latter encompassed all cortical layers. To quantify the distribution of GABA+ and PV+ immunoreactivity in control tissue (i.e., the data shown in *Figure 2*, and the 'OUT' data in *Figure 7*), we counted single- and double-labeled cells across two 100-μm-wide ROIs extending through all layers in each tissue section for a total of six ROIs across three sections. The ROIs for this analysis were selected to be millimeters away from the V1 region containing the viral injection sites. Cell counting was performed by two undergraduate researchers (AI, PB) and reviewed for accuracy by senior lab members (FF, AA). Cortical layer boundaries were determined using DAPI staining or PV-IHC (after confirming the layer boundaries based on PV-IHC matched those seen in DAPI). Data collected in Neurolucida were exported to Excel (Microsoft) and SPSS (IBM) software for quantitative and statistical analyses.

## Statistical analysis

To compare cell counts and neuronal densities across different viral serotypes (for the GABA-AAVs), different viral volumes (for the PV-AAV), or different cortical layers, we used an ANOVA, when the data were normally distributed, and either the non-parametric independent-samples Kruskal–Wallis test, an independent-samples median test, or the Mann–Whitney *U* test for data that were not normally distributed, unless otherwise indicated in the 'Results' section. All multiple comparisons were Bonferroni-corrected.

## Materials availability statement

All materials used in this study are available commercially except for the AAV-h56D-tdTomato virus, which can be obtained directly from the Zemelman laboratory (UT Austin).

## Acknowledgements

We thank Kesi Sainsbury for histological assistance. This work was supported primarily by a grant from the National Institute of Health (NIH) to AA (R01 EY031959). Other grants were provided by the NIH (R01 EY026812), the National Science Foundation (IOS 1755431), and the Mary Boesche endowed Professorship, to AA; an unrestricted grant from Research to Prevent Blindness, Inc and a core grant from the NIH (P30 EY014800) to the Department of Ophthalmology, University of Utah.

## Additional information

### Funding

| Funder | Grant reference number | Author |
| --- | --- | --- |
| National Eye Institute | R01 EY031959 | Alessandra Angelucci |
| National Eye Institute | R01 EY026812 | Alessandra Angelucci |
| National Science Foundation | IOS 1755431 | Alessandra Angelucci |
| Research to Prevent Blindness | unrestricted grant to the Dept of Ophthalmology | Alessandra Angelucci |
| National Eye Institute | P30 EY014800 | Alessandra Angelucci |
| Research to Prevent Blindness | Univ of Utah | Alessandra Angelucci |

The funders had no role in study design, data collection and interpretation, or the decision to submit the work for publication.

### Author contributions

Frederick Federer, Conceptualization, Data curation, Formal analysis, Investigation, Methodology, Project administration, Supervision, Validation, Visualization, Writing – original draft, Writing – review and editing; Justin Balsor, Conceptualization, Formal analysis, Investigation, Methodology; Alexander Ingold, Data curation, Investigation; David P Babcock, Data curation; Jordane Dimidschstein, Resources, Methodology, Writing – review and editing; Alessandra Angelucci, Formal analysis, Conceptualization, Supervision, Funding acquisition, Investigation, Visualization, Methodology, Writing – original draft, Project administration, Writing – review and editing

### Author ORCIDs

Frederick Federer http://orcid.org/0000-0002-1340-865X
Alexander Ingold https://orcid.org/0009-0007-5238-0016
Alessandra Angelucci https://orcid.org/0000-0002-1957-2231

### Ethics

All procedures were approved by the University of Utah Institutional Animal Care and Use Committee (IAUC protocol No. 21-12015) and conformed to the ethical guidelines set forth by the USDA and NIH.

Reviewer #1 (Public Review): https://doi.org/10.7554/eLife.97673.3.sa1
Reviewer #3 (Public Review): https://doi.org/10.7554/eLife.97673.3.sa2
Author response https://doi.org/10.7554/eLife.97673.3.sa3

## Additional files

### Supplementary files

• Supplementary file 1. File 1 reports the injection parameters used for each AAV-h56D and AAV-PHP.eB-S5E2 injection case.

• Supplementary file 2. File 2 reports the specificity and coverage for each individual AAV-PHP.eB-S5E2-tdT injection case.

• MDAR checklist

### Data availability

Source data for the figures can be found on DRYAD at https://doi.org/10.5061/dryad.ht76hdrr3.

The following dataset was generated:

| Author(s) | Year | Dataset title | Dataset URL | Database and Identifier |
|---|---|---|---|---|
| Federer F, Angelucci A | 2024 | Laminar specificity and coverage of viral-mediated gene expression restricted to GABAergic interneurons and their parvalbumin subclass in marmoset primary visual cortex | https://doi.org/10.5061/dryad.ht76hdrr3 | Dryad Digital Repository, 10.5061/dryad.ht76hdrr3 |

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
