## [Editor Report · eLife assessment]

Unlocking the potential of molecular genetic tools (optogenetics, chemogenetics, sensors, etc.) for the study of systems neuroscience in nonhuman primates requires the development of effective regulatory elements for cell-type-specific expression to facilitate circuit dissection. This study provides a **valuable** building block by carefully characterizing the laminar expression profile of two optogenetic enhancers, one designed for general GABA+ergic neurons (h56D) and the second (S5E2) for parvalbumin+ cell-type selective expression in the marmoset primary visual cortex. This study contributes **solid** evidence to our understanding of these tools but is limited by the understandably small number of animals used.

---

## [Referee Report · Reviewer #1 (Public Review)]

Summary:

Federer et al. tested AAVs designed to target GABAergic cells and parvalbumin-expressing cells in marmoset V1. Several new results were obtained. First, AAV-h56D targeted GABAergic cells with high specificity but in ways that varied across serotype and layer. Second, AAV-PHP.eB.S5E2 targeted parvalbumin-expressing neurons with similarly high specificity. Third, immunohistochemical GABA and PV signals were attenuated near viral injection sites.

A strength of this study is the analysis of marker gene expression at AAV injection sites. Some endogenous genes are difficult to detect following AAV injections, which is an important observation. A second contribution is the demonstration that AAV-S5E2 drives transgene expression selectively in parvalbumin-expressing neurons when vectors are delivered intraparenchymally (the study introducing AAV-S5E2 used intravenous injections).

A weakness of this study is that the data set is small. Which of the results would hold up had a larger number of injections been made into a larger number of marmosets remains unclear.

A major goal of this study was to quantify the specificity and coverage of AAV-h56D and AAV-S5E2 vectors in marmoset cortex. This goal was achieved. This report provides a valuable guide for other investigators using these tools. It also provides a rigorous survey of the laminar distributions of GABA+ and PV+ neurons in marmoset V1 which has value independent of the viral injections.

---

## [Referee Report · Reviewer #3 (Public Review)]

Summary:

Federer et al. describe the laminar profiles of GABA+ and of PV+ neurons in marmoset V1. They also report on the selectivity and efficiency of expression of a PV-selective enhancer (S5E2) across laminae. Three further viruses were tested, with a view to characterizing the expression profiles of a GABA-selective enhancer (h56d), but these results are preliminary.

Strengths:

The derivation of cell-type specific enhancers is key for translating the types of circuit analyses that can be performed in mice - which rely on germline modifications for access to cell-type specific manipulation - in higher-order mammals. Federer et al. further validate the utility of S5E2 as a PV-selective enhancer in NHPs.

Additionally, the authors characterize the laminar distribution pattern of GABA+ and PV+ cells in V1. This survey may prove valuable to researchers seeking to understand and manipulate the microcircuitry mediating the excitation-inhibition balance in this region of the marmoset brain.

Weaknesses:

Enhancer/promoter specificity and efficiency cannot be directly compared, because they were packaged in different serotypes of AAV.

The three different serotypes of AAV expressing reporter under the h56D promoter were only tested once each, and all in the same animal. There are many variables that can contribute to the success (or failure) of a viral injection, so observations with an n = 1 cannot be considered reliable.

Added after revision:

The revisions satisfy my concerns. In particular, the new language to qualify the strength of evidence relating to the interpretation of the data relating to the 3 different serotypes of virus used to test h56D is appropriate.

---

## [Author Response]

The following is the authors’ response to the original reviews.

**Public Reviews**

**Reviewer #1 (Public Review):**
Summary:Federer et al. tested AAVs designed to target GABAergic cells and parvalbumin-expressing cells in marmoset V1. Several new results were obtained. First, AAV-h56D targeted GABAergic cells with >90% specificity, and this varied with serotype and layer. Second, AAV-PHP.eB.S5E2 targeted parvalbumin-expressing neurons with up to 98% specificity. Third, the immunohistochemical detection of GABA and PV was attenuated near viral injection sites.Strengths:Vormstein-Schneider et al. (2020) tested their AAV-S5E2 vector in marmosets by intravenous injection. The data presented in this manuscript are valuable in part because they show the transduction pattern produced by intraparenchymal injections, which are more conventional and efficient.

Our manuscript additionally provides detailed information on the laminar specificity and coverage of these viral vectors, which was not investigated in the original studies.

Weaknesses:The conclusions regarding the effects of serotype are based on data from single injection tracks in a single animal. I understand that ethical and financial constraints preclude high throughput testing, but these limitations do not change what can be inferred from the measurements. The text asserts that "...serotype 9 is a better choice when high specificity and coverage across all layers are required". The data presented are consistent with this idea but do not make a strong case for it.

We are aware of the limitations of our results on the AAV-h56D. We agree with the Reviewer that a single injection per serotype does not allow us to make strong statements about differences between the 3 serotypes. Therefore, in the revised version of the manuscript we have tempered our claims about such differences and use more caution in the interpretation of these data (Results p. 6 and Discussion p.10). Despite this weakness, we feel that these data still demonstrate high efficiency and specificity across cortical layers of transgene expression in GABA cells using the h56D promoter, at least with two of the 3 AAV serotypes we tested. We feel that in itself this is sufficiently useful information for the primate community, worthy of being reported. Due to cost, time and ethical considerations related to the use of primates, we chose not to perform additional experiments to determine precise differences among serotypes. Thus, for example, while it is possible that had we replicated these experiments, serotype 7 could have proven equally efficient and specific as the other two serotypes, we felt answering this question did not warrant additional experiments in this precious species.

A related criticism extends to the analysis of Injection volume on viral specificity. Some replication was performed here, but reliability across injections was not reported. My understanding is that individual ROIs were treated as independent observations. These are not biological replicates (arguably, neither are multiple injection tracks in a single animal, but they are certainly closer). Idiosyncrasies between animals or injections (e.g., if one injection happened to hit one layer more than another) could have substantial impacts on the measurements. It remains unclear which results regarding injection volume or serotype would hold up had a large number of injections been made into a large number of marmosets.

For the AAV-S5E2, we made a total of 7 injections (at least 2 at each volume), all of which, irrespective of volume, resulted in high specificity and efficiency for PV interneurons. Our conclusion is that larger volumes are slightly less specific, but the differences are minimal and do not warrant additional injections. Additionally, we kept all the other parameters across animals constant (see new Supplementary Table 1), all of our injections involved all cortical layers, and the ROIs we selected for counts encompassed reporter protein expression across all layers. To provide a better sense of the reliability of the results across injections, in the revised version of the manuscript we now provide results for each of the AAV-S5E2 injection case separately in a new Supplementary Table 2. The results in this table indicate the results are indeed rather consistent across cases with slightly greater specificity for injection volumes in the range of 105-180 nl.

**Reviewer #2 (Public Review):**
This is a straightforward manuscript assessing the specificity and efficiency of transgene expression in marmoset primary visual cortex (V1), for 4 different AAV vectors known to target transgene expression to either inhibitory cortical neurons (3 serotypes of AAV-h56D-tdTomato) or parvalbumin (PV)+ inhibitory cortical neurons in mice. Vectors are injected into the marmoset cortex and then postmortem tissue is analyzed following antibody labeling against GABA and PV. It is reported that: "in marmoset V1 AAV-h56D induces transgene expression in GABAergic cells with up to 91-94% specificity and 80% efficiency, depending on viral serotype and cortical layer. AAV-PHP.eB-S5E2 induces transgene expression in PV cells across all cortical layers with up to 98% specificity and 86-90% efficiency."These claims are largely supported but slightly exaggerated relative to the actual values in the results presented. In particular, the overall efficiency for the best h56D vectors described in the results is: "Overall, across all layers, AAV9 and AAV1 showed significantly higher coverage (66.1{plus minus}3.9 and 64.9%{plus minus}3.7)". The highest coverage observed is just in middle layers and is also less than 80%: "(AAV9: 78.5%{plus minus}9.1; AAV1: 76.9%{plus minus}7.4)".

In the abstract, we indeed summarize the overall data and round up the decimals, and state that these percentages are upper bound but that they vary by serotype and layer while in the Results we report the detailed counts with decimals. To clarify this, in the revised version of the Abstract we have changed 80% to 79% and emphasize even more clearly the dependence on serotype and layer. We have amended this sentence of the Abstract as follows: “We show that in marmoset V1 AAV-h56D induces transgene expression in GABAergic cells with up to 91-94% specificity and 79% efficiency, but this depends on viral serotype and cortical layer.”

For the AAV-PHP.eB-S5E2 the efficiency reported in the abstract (“86-90%) is also slightly exaggerated relative to the results: “Overall, across all layers coverage ranged from 78%{plus minus}1.9 for injection volumes >300nl to 81.6%{plus minus}1.8 for injection volumes of 100nl.”

Indeed, the numbers in the Abstract are upper bounds, for example efficiency in L4A/B with S5E2 reaches 90%. To further clarify this important point, in the revised abstract we now state ”AAV-PHP.eB-S5E2 induces transgene expression in PV cells across all cortical layers with up to 98% specificity and 86-90% efficiency, depending on layer”.

These data will be useful to others who might be interested in targeting transgene expression in these cell types in monkeys. Suggestions for improvement are to include more details about the vectors injected and to delete some comments about results that are not documented based on vectors that are not described (see below).Major comments:Details provided about the AAV vectors used with the h56D enhancer are not sufficient to allow assessment of their potential utility relative to the results presented. All that is provided is: "The fourth animal received 3 injections, each of a different AAV serotype (1, 7, and 9) of the AAV-h56D-tdTomato (Mehta et al., 2019), obtained from the Zemelman laboratory (UT Austin)." At a minimum, it is necessary to provide the titers of each of the vectors. It would also be helpful to provide more information about viral preparation for both these vectors and the AAVPHP.eB-S5E2.tdTomato. Notably, what purification methods were used, and what specific methods were used to measure the titers?

We thank the Reviewer for this comment. In the revised version of the manuscript, we now provide a new Supplementary Table 1 with titers and other information for each viral vector injection. We also provide information regarding viral preparation in a new sections in the Methods entitled “ Viral Preparation” (p12).

The first paragraph of the results includes brief anecdotal claims without any data to support them and without any details about the relevant vectors that would allow any data that might have been collected to be critically assessed. These statements should be deleted. Specifically, delete: “as well as 3 different kinds of PV-specific AAVs, specifically a mixture of AAV1-PaqR4-Flp and AAV1-h56D-mCherry-FRT (Mehta et al., 2019), an AAV1-PV1-ChR2-eYFP (donated by G. Horwitz, University of Washington),” and delete “Here we report results only from those vectors that were deemed to be most promising for use in primate cortex, based on infectivity and specificity. These were the 3 serotypes of the GABA-specific pAAV-h56D-tdTomato, and the PV-specific AAVPHP.eB-S5E2.tdTomato.” These tools might in fact be just as useful or even better than what is actually tested and reported here, but maybe the viral titer was too low to expect any expression.

These data are indeed anecdotal, but we felt this could be useful information, potentially preventing other primate labs from wasting resources, animals and time, particularly, as some of these vectors have been reported to be selective and efficient in primate cortex, which we have not been able to confirm. We made several injections in several animals of those vectors that failed either to infect a sufficient number of cells or turned out to be poorly specific. Therefore, the negative results have been consistent in our hands. But we agree with the Reviewer that our negative results could have depended on factors such as titer. In the revised version of the manuscript, following the reviewer’s suggestion, we have deleted this information.

Based on the description in the Methods it seems that no antibody labeling against TdTomato was used to amplify the detection of the transgenes expressed from the AAV vectors. It should be verified that this is the case - a statement could be added to the Methods.

That is indeed the case. We used no immunohistochemistry to enhance the reporter proteins as this was unnecessary. The native/ non-amplified tdT signal was strong. This is now stated in the methods (p.12).

**Reviewer #3 (Public Review):**
Summary:Federer et al. describe the laminar profiles of GABA+ and of PV+ neurons in marmoset V1. They also report on the selectivity and efficiency of expression of a PV-selective enhancer (S5E2). Three further viruses were tested, with a view to characterizing the expression profiles of a GABA-selective enhancer (h56d), but these results are preliminary.Strengths:The derivation of cell-type specific enhancers is key for translating the types of circuit analyses that can be performed in mice - which rely on germline modifications for access to cell-type specific manipulation - in higher-order mammals. Federer et al. further validate the utility of S5E2 as a PV-selective enhancer in NHPs.Additionally, the authors characterize the laminar distribution pattern of GABA+ and PV+ cells in V1. This survey may prove valuable to researchers seeking to understand and manipulate the microcircuitry mediating the excitation-inhibition balance in this region of the marmoset brain.Weaknesses:Enhancer/promoter specificity and efficiency cannot be directly compared, because they were packaged in different serotypes of AAV.The three different serotypes of AAV expressing reporter under the h56D promoter were only tested once each, and all in the same animal. There are many variables that can contribute to the success (or failure) of a viral injection, so observations with an n = 1 cannot be considered reliable.

This is an important point that was also brough up by Reviewer 1, which we have addressed in our reply-to-Reviewer 1. For clarity and convenience, below we copy our response to Reviewer 1.

“We are aware of the limitations of our results on the AAV-h56D. We agree with the Reviewer that a single injection per serotype does not allow us to make strong statements about differences between the 3 serotypes. Therefore, in the revised version of the manuscript we will temper our claims about such differences and use more caution in the interpretation of these data. Despite this weakness, we feel that these data still demonstrate high efficiency and specificity across cortical layers of transgene expression in GABA cells using the h56D promoter, at least with two of the 3 AAV serotypes we tested. We feel that in itself this is sufficiently useful information for the primate community, worthy of being reported. Due to cost, time and ethical considerations related to the use of primates, we chose not to perform additional experiments to determine precise differences among serotypes. Thus, for example, while it is possible that had we replicated these experiments, serotype 7 would have proven equally efficient and specific as the other two serotypes, we felt answering this question did not warrant additional experiments in this precious species.”

The language used throughout conflates the cell-type specificity conferred by the regulatory elements with that conferred by the serotype of the virus.

Authors’ reply. In the revised version of the manuscript, we have corrected ambiguous language throughout.

**Recommendations for the authors**

**Reviewer #1 (Recommendations For The Authors):**
My Public Review comments can be addressed by dialing down the interpretation of the data or providing appropriate caveats in the presentation of the relevant results and their discussion.

We have done so. See text additions on p. 6 of the Results and p.10 of the Discussion.

Minor comments:92% of PV+ neurons in the marmoset cortex were GABAergic. Can the authors speculate on the identity of the 8% PV+/GABA- neurons (e.g., on the basis of morphology)? Are they likely excitatory? Are they more likely to represent failures of GABA staining?

We do not know what the other 8% of PV+/GABA- neurons are because we did not perform any other kind of IHC staining. Our best guess is that at least to some extent these represent failures of GABA staining, which is always challenging to perform in primate cortex. However, in mouse PV expression has been demonstrated in a minority of excitatory neurons.

"Coverage of the PV-AAV was high, did not depend on injection volume.." The fact that the coverage did not depend on injection volume presumably depends, at least in part, on how ROIs were selected. Surely different volumes of injection transduce different numbers of neurons at different distances from the injection track. This should be clarified.

The ROIs were selected at the center of the injected site/expression core from sections in which the expression region encompassed all cortical layers. Of course, larger volumes of injection resulted in larger transduced regions and therefore overall larger number of transduced neurons, but we counted cells only withing 100 µm wide ROIs at the center of the injection and the percent of transduced PV cells in this core region did not vary significantly across volumes. We have clarified the methods of ROI selection (see Methods pp. 13).

Figure 2. What is meant by “absolute” in the legend for Figure 2? (How does “mean absolute density” differ from “mean density?”)

We meant not relative, but this is obvious from the units, so we have removed the word “absolute” in the legend.

Some non-significant p-values are indicated by "p>0.05" whereas others are given precisely (e.g., p = 1). Please provide precise p-values throughout. Also, the p-value from a surprisingly large number of comparisons in the first section of the results is "1". Is this due to rounding? Is it possible to get significance in a Bonferroni-corrected Kruskal-Wallis test with only 6 observations per condition?

We now report exact p values throughout the manuscript (with a couple of exceptions where, in order to avoid reporting a large number of p values which interrupts the flow of the manuscript) we provide the upper bound value and state all those comparisons were below that value. The minimum sample size for Kruskal Wallis is 5, for each group being compared, and we our sample is 6 per group.

Figure 3: The density of tdTomato-expressing cells appears to be greater at the AAV9 injection site than at the AAV1 injection site in the example sections shown. Might some of the differences between serotypes be due to this difference? I would imagine that resolving individual cells with certainty becomes more difficult as the amount of tdTomato expression increases.

There was an error in the scale bar of Fig. 3C, so that the AAV1 injection site was shown at higher magnification than indicated by the wrong scale bar. Hence the density of tdTomato appeared lower than it is. Moreover, the tdT expression region shown in Fig. 3A is a merge of two sections, while it is only from a single section in panels B and C, leading to the impression of higher density of infected cells in panel A. The pipette used for the injection in panel A was not inserted perfectly vertical to the cortical surface, resulting in an injection site that did not span all layers in a single section; thus, to demonstrate that the injection indeed encompassed all layers (and that the virus infected cells in all layers), we collapsed label from two sections. We have now corrected the magnification of panel C so that it matches the scale bar in panel A, and specify in the figure legend that panel A label is from two sections.

Text regarding Figure 3: The term “injection sizes” is confusing. I think it is intended to mean “the area over which tdTomato-expressing cells were found” but this should be clarified.

Throughout the manuscript, we have changed the term injection site to “viral-expression region”.

Figure 3: What were the titers of the three AAV-h56D vectors?

Titers are now reported in the new Supplementary Table 1.

Figure 3: The yellow box in Figure 3C is slightly larger than the yellow boxes in 3A and 3B. Is this an error or should the inset of Figure 3 have a scale bar that differs from the 50 µm scale bar in 3A?

There were indeed errors in scale bars in this figure, which we have now corrected. Now all boxes have the same scale bar.

Was MM423 one of the animals that received the AAV-h56D injections or one of the three that received AAV-S5E2 injection?

This is an animal that received a 315nl injection of AAV-PHP.eB-S5E2.tdTomato. This is now specified in the Methods (see p. 12) and in the new Supplementary Table 1.

Please provide raw cell counts and post-injection survival times for each animal.

We now provide this information in Supplementary Tables 1 and 2.

How were the different injection volumes of the AAV-S5E2 virus arranged by animal? Which volume of the AAV-S5E2 virus was injected into the two animals who received single injections?

We now provide this information in Supplementary Table 1.

Figure 6A: the point is made in the text that "[the distribution of tdT+ and PV+ neurons] did not differ significantly... peaking in L2/3 and 4C " Is the fact that the number of tdT+ and PV+ peak in layers 2/3 and 4C a consequence of these layers being thicker than the others? If so, this statement seems trivial.

No, and this is the reason why we measured density in addition to percent of cells across layers in Figure 2. Figure 2B shows that even when measuring density, therefore normalizing by area, GABA+ and PV+ cell density still peaks in L2/3 and 4. Thus, these peaks do not simply reflect the greater thickness of these layers.

Do the authors have permission to use data from Xu et al. 2010?

Yes, we do.

**Reviewer #2 (Recommendations For The Authors):**
Minor comments:"Viral strategies to restrict gene expression to PV neurons have also been recently developed (Mehta et al., 2019; Vormstein-Schneider et al., 2020)." Mich et al. should also be cited here. Cell Rep. 2021;34(13):108754.

We thank the reviewer for pointing out this missing references. This is now cited.

“GABA density in L4C did not differ from any other layers, but the percent of GABA+ cells in L4C was significantly higher than in L1 (p=0.009) and 4A/B (p=<0.0001).” This and other similar observations depend on calculating the percentage of cells relative to the total number of DAPI-labeled cells in each layer. Since it is apparent that there must be considerable variability between layers, it would be helpful to add a histogram showing the densities of all DAPI-labeled cells for each layer.

This is not how we calculated density. Density, as now clarified in the Results on p. 4, was defined as the number of cells per unit area. Counts in each layer were divided by each layers’ counting area. This corrects for differences in number of total labeled cells per layer. Therefore, reporting DAPI density is not necessary (we did not count DAPI cell density per layer).

"Identical injection volumes of each serotype, delivered at 3 different cortical depths (see Methods), resulted in different injection sizes, suggesting the different serotypes have different capacity of infecting cortical neurons. AAV7 produced the smallest injection site, which additionally was biased to the superficial and deep layers, with only few cells expressing tdT in the middle layers (Fig. 3B). AAV9 (Fig. 3A) and AAV1 (Fig. 3C) resulted in larger injection sites and infected all cortical layers." Differences noted here might reflect either differences related to the AAV serotype or to differences in titers. Please add details about titers for each vector and add comments as appropriate. Another interpretation would be that there are differences in viral spread within the tissue.

We have now added Supplementary Table 1 which reports titers in addition to other information about injections. The titers and volumes used for AAV9 and AAV7 were identical, while the titer for AAV1 was higher. Therefore, the differences in infectivity, particularly the much smaller expression region obtained with AAV7 cannot be attributed to titer. Likely this is due to differences in tropism and/or viral spread among serotypes. This is now discussed (see Results p. 5bottom and 6 top).

“Recently, several viral vectors have been identified that selectively and efficiently restrict gene expression to GABAergic neurons and their subtypes across several species, but a thorough validation and characterization of these vectors in primate cortex has lacked.” Is this really a fair statement, or is the characterization presented here also lacking? Methods used by others for quantifying specificity and efficiency are essentially the same as used here. See for example Mich et al. (which is not cited).

The original validation in primates of the vectors examined in our study was based on small tissue samples and did not examine the laminar expression profile of transgene expression induced by these enhancer-AAVs. For example, the validation of the h56D-AAV in marmoset cortex in the original paper by Mehta et al (2019) was performed on a tissue biopsy with no knowledge of which cortical layers were included in the tissue sample. The only study that shows laminar expression in primate cortex (Mich et al., which is now cited), only shows qualitative images of viral expression across layers, reporting total specificity and coverage pooled across samples; moreover, the study by Mich et al. deals with different PV-specific enhancers than the ones characterized in our study. Unlike any of the previous studies, here we have quantified specificity and coverage across layers.

"Specifically, we have shown that the GABA-specific AAV9-h56D (Mehta et al., 2019) induces transgene expression in GABAergic cells with up to 91-94% specificity and 80% coverage, and the PV-specific AAV-PHP.eB-S5E2 (Vormstein-Schneider et al., 2020) induces transgene expression in PV cells with up to 98% specificity and 86-90% coverage." These statements in the discussion repeat the somewhat exaggerated coverage numbers noted above for the Abstract.

The averages across all layers are reported in the Results. The Discussion, abstract and discussion report upper limits, and this is made clear by stating “up to”, and now we have also added “depending on layer”.

**Reviewer #3 (Recommendations For The Authors):**
Abstract:• Ln 2: Can you be more specific about what you mean by the 'various functions of inhibition'? e.g. do you mean 'the various inhibitory influences on the local microcircuit' or similar?

These are listed in the introduction to the paper but there is no space in the abstract to do so. Now the sentence reads: “various computational functions of…”.

• Ln 5: 'has' to 'is'/'has been'.

The grammar here is correct “has derived”.

• Ln 6: humans are primates! Maybe change this to 'nonhuman primates'?

We have added “non-human”

• Ln n-1: 'viral vectors represent' -> 'viral vectors are'.

We have changed it to “are”

Intro:• Many readers may expect 'VIP' to be listed as the third major sub-class of interneurons. Could you note that the 5HT3a receptor-expressing group includes VIP cells?

Done (p.3).

• "Understanding cortical inhibitory neuron function in the primate is critical for understanding cortical function and dysfunction in the model system closest to humans" - this seems close to being circular logic (not quite, but close). Could you modify this sentence to reflect why understanding cortical function and dysfunction in NHP may be of interest?

This sentence now reads (p.3):” Understanding cortical inhibitory neuron function in the primate is critical for understanding cortical function and dysfunction in the model system closest to humans, where cortical inhibitory neuron dysfunction has been implicated in many neurological and psychiatric disorders, such as epilepsy, schizophrenia and Alzheimer’s disease (Cheah et al., 2012; Verret et al., 2012; Mukherjee et al., 2019)”. We also note that this was already stated in the previous version of the paper but in the Discussion section which read (and still reads on p. 9 2nd paragraph): “It is important to study inhibitory neuron function in the primate, because it is unclear whether findings in mice apply to higher species, and inhibitory neuron dysfunction in humans has been implicated in several neurological and psychiatric disorders (Marin, 2012; Goldberg and Coulter, 2013; Lewis, 2014).”.

• "In particular, two recent studies have developed recombinant adeno-associated viral vectors (AAV) that restrict gene expression to GABAergic neurons". This sentence places the emphasis on the wrong component of the technology. The fact that AAV was used is irrelevant; these constructs could equally have been packaged in a lenti, CAV, HSV, rabies, etc. The emphasis should be on the recently developed regulatory elements (the enhancers/promoters).Same problem with the following excerpts; this text implies that the serotype/vector confers cell-type selectivity, but the results presented do not support this assertion (the promoter/enhancer is what confers the selectivity).• "specifically, three serotypes of an AAV that restricts gene expression to GABAergic neurons".• "one serotype of an AAV that restricts gene expression to PV cells".• "GABA- and PV-specific AAVs".• "GABA-specific AAV" (in results).• "PV-specific AAVs".• "In this study, we have characterized several AAV vectors designed to restrict expression to GABAergic cells" (in discussion).• "GABA-virus". GABA is a NT, not a virus.

We have modified the language in all these sections and throughout the manuscript.

Results:• Enhancer specificity and efficiency cannot be directly compared, because they were packaged in different serotypes of AAV.

We agree, and in fact we are not making comparisons between different enhancers (i.e., S5E2 and h56D).

The three different serotypes of AAV expressing reporter under the h56D promoter were only tested once each, and all in the same animal. There are many variables that can contribute to the success (or failure) of a viral injection, so observations with an n = 1 cannot be considered reliable.The authors need to either: (1) replicate the h56D virus injections in (at least) a second animal, or (2) rewrite the paper to focus on the AAV.PhP mDlx virus alone - for which they have adequate data - and mention the h56D data as an anecdotal result, with clear warnings about the preliminary nature of the observations due to lack of replication.

We agree about the lack of sufficient data to make strong statements about the differences between serotypes for the h56D-AAV. In the revised version of the manuscript, following the Reviewers’ suggestion, we have chosen to temper our claims about differences between serotypes for the h56D enhancer and use more caution in the interpretation of these data. We feel that these data still demonstrate sufficiently high efficiency and specificity across cortical layers of transgene expression in GABA cells using the h56D promoter, at least with two of the 3 AAV serotypes we tested, to warrant their use in primates. Due to cost, time and ethical considerations related to the use of primates, we chose not to perform additional experiments to determine precise differences among serotypes. Thus, for example, while it is possible that had we replicated these experiments, serotype 7 could have proven equally efficient and specific as the other two serotypes, we felt answering this question did not warrant additional experiments in this precious species. Our edits in regard to this point can be found in the Results on p. 6 and Discussion on p. 10.

• Did the authors compare h56D vs mDlx? This would be a useful and interesting comparison.

We did not.

• 3 tissue sections were used for analysis. How were these selected? Did the authors use a stereological approach?

For the analysis in Fig. 2, the 3 sections were randomly selected and for the positioning of the ROIs we selected a region in dorsal V1 anterior to the posterior pole (to avoid laminar distortions due to the curvature of the brain). This is now specified (see p. 4).

• "both GABA+ and PV+ cells peak in layers" revise for clarity (e.g., the counts peak).

In now reads “GABA+ and PV+ cell percent and density” (see p.4).

• "we refer to this virus as GABA-AAV" these are 3 different viruses!

The idea here was to use an abbreviation instead of using the full viral name every single time. Clearly the reviewer does not like this, so we have removed this convention throughout the paper and now specify the entire viral name each time.

• "Identical injection volumes of each serotype, delivered at 3 different cortical depths (see Methods), resulted in different injection sizes". Do you mean 'resulted in different volumes of expression'?

Yes. We have now rephrased this as follows: “…resulted in viral expression regions that differed in both size as well as laminar distribution” (p.5).

• “suggesting the different serotypes have different capacity of infecting cortical neurons”. You can’t draw any firm conclusions from a single injection. The rest of this section of the results, along with the whole of Figure 4, and Figure 7a-d, is in danger of being misleading. Please remove. The best you can do here is to say ‘we injected 3 different viruses that express reporter under the h56D promoter. The results are shown in Figure 3, but these are anecdotal, as only a single injection of each virus was performed’. You could then note in the discussion to what extent these results are consistent with the existing literature (e.g., AAV9 often produces good coverage in NHP – anterograde and retrograde, AAV1 also works well in the CNS, although generally doesn’t infect as aggressively as AAV9. I’m not familiar with any attempts to use AAV7).

With respect to Fig. 4, our approach in the revised version is detailed above. For convenience we copy it below here. With respect to Fig 7A-D, we feel the results are more robust as the data from the 3 serotypes here were pooled together, as the 3 serotype similarly downregulated GABA and PV expression at the injection site, and we do not make any statement about differences among serotypes for the data shown in Fig. 7A-D.

“In the revised version of the manuscript, following the Reviewer ’s suggestion, we have chosen to temper our claims about differences between serotypes for the h56D enhancer and use more caution in the interpretation of these data (see revised text in the Results on p. 6 and in the Discussion on p. 10). We feel that these data still demonstrate sufficiently high efficiency and specificity across cortical layers of transgene expression in GABA cells using the h56D promoter, at least with two of the 3 AAV serotypes we tested, to warrant their use in primates. Due to cost, time and ethical considerations related to the use of primates, we chose not to perform additional experiments to determine precise differences among serotypes. Thus, for example, while it is possible that had we replicated these experiments, serotype 7 could have proven equally efficient and specific as the other two serotypes, we felt answering this question did not warrant additional experiments in this precious species.”

• Figure 3: why the large variation in tissue quality? Are the 3 upper images taken at the same magnification? If not, they need different scale bars. The cells in A (upper row) look much smaller than those in B and C, and the size of the 'inset' box varies.

We thank the reviewer for noticing this. We discovered an error in the scale bar of Fig. 3C, so that the AAV1 injection site was shown at higher magnification than indicated by the wrong scale bar. We have now corrected the error in scale bars. We have also fixed the different box sizes.

• "Overall, across all layers coverage ranged from 78%{plus minus}1.9 for injection volumes >300nl to 81.6%{plus minus}1.8 for injection volumes of 100nl." Coverage didn't differ between layers, so revise this to: "Overall, across all layers coverage ranged from 78% to 81.6%." or give an overall mean (~80%).

We have corrected the sentence as suggested by the Reviewer (see p. 8 first paragraph).

• "extending farther from the borders" -> "extending beyond the borders".

We have corrected the sentence as suggested by the Reviewer (see p. 8).

• "The reduced GABA and PV immunoreactivity caused by the viruses implies that the specificity of the viruses we have validated in this study is likely higher than estimated". Yes, but for balance you should also note that they may harm the physiology of the cell.

We have added a sentence acknowledging this to the Discussion. Specifically, on p. 10, we now state: “However, this reduced immunoreactivity raises concerns about the virus or high levels of reporter protein possibly harming the cell physiology.”

Discussion:• "but a thorough validation and characterization of these vectors in primate cortex has lacked" better to say "has been limited", because Dimidschstein 2016 (marmoset V1) and Vormstein-schneider 2020 (macaque S1 and PFC) both reported expression in NHP.

We have added the following sentence to this paragraph of the Discussion. “In particular, previous studies have not characterized the specificity and coverage of these vectors across cortical layers.”(see p. 8).

• "whether finding in mice" -> 'whether findings in mice'.

Corrected, thanks.

• The discussion re: species differences is missing reference to Kreinen 2020 (10.1038/s41586-020-2781-z).

This reference has been added. Thanks.

• “Injections of about 200nl volume resulted in higher specificity (95% across layers) and coverage” – this is misleading. The coverage was not statistically different among injection volumes.

We have added the following sentence: ”although coverage did not differ significantly across volumes.” (see p. 10).

• "it is possible that subtle alteration of the cortical circuit upon parenchymal injection of viruses (including AAVs) leads to alteration of activity-dependent expression of PV and GABA." Or (and I would argue, more likely) the expression of large quantities of your big reporter protein compromised the function of the cell, leading to reduced expression of native proteins. You don't mention any IHC to amplify the RFP signal, so I'm assuming that your images are of direct expression. If so, you are expressing A LOT of reporter protein.

We have added a sentence acknowledging this to the Discussion. Specifically, on p. 10, we now state: “However, this reduced immunoreactivity raises concerns about the virus or high levels of reporter protein possibly harming the cell physiology.”

Methods:• It's difficult to piece together which viruses were injected in which monkeys, at what volumes, and at what titer. Please compile this info into a table for ease of reference (including any other relevant parameters).

We now provide a Supplementary Table 1.